# A Time-of-Flight Image Sensor Using 8-Tap P-N Junction Demodulator Pixels

**DOI:** 10.3390/s23083987

**Published:** 2023-04-14

**Authors:** Ryosuke Miyazawa, Yuya Shirakawa, Kamel Mars, Keita Yasutomi, Keiichiro Kagawa, Satoshi Aoyama, Shoji Kawahito

**Affiliations:** 1Graduate School of Integrated Science and Technology, Shizuoka University, Hamamatsu 432-8011, Japan; 2Research Institute of Electronics, Shizuoka University, Hamamatsu 432-8011, Japan; 3Brookman Technology, Inc., Hamamatsu 430-0936, Japan

**Keywords:** image sensor, PN-junction demodulator (PND), time-of-flight (ToF), depth-adaptive time-gating-number assignment (DATA), depth precision

## Abstract

This paper presents a time-of-flight image sensor based on 8-Tap P-N junction demodulator (PND) pixels, which is designed for hybrid-type short-pulse (SP)-based ToF measurements under strong ambient light. The 8-tap demodulator implemented with multiple p-n junctions used for modulating the electric potential to transfer photoelectrons to eight charge-sensing nodes and charge drains has an advantage of high-speed demodulation in large photosensitive areas. The ToF image sensor implemented using 0.11 µm CIS technology, consisting of an 120 (H) × 60 (V) image array of the 8-tap PND pixels, successfully works with eight consecutive time-gating windows with the gating width of 10 ns and demonstrates for the first time that long-range (>10 m) ToF measurements under high ambient light are realized using single-frame signals only, which is essential for motion-artifact-free ToF measurements. This paper also presents an improved depth-adaptive time-gating-number assignment (DATA) technique for extending the depth range while having ambient-light canceling capability and a nonlinearity error correction technique. By applying these techniques to the implemented image sensor chip, hybrid-type single-frame ToF measurements with depth precision of maximally 16.4 cm (1.4% of the maximum range) and the maximum non-linearity error of 0.6% for the full-scale depth range of 1.0–11.5 m and operations under direct-sunlight-level ambient light (80 klux) have been realized. The depth linearity achieved in this work is 2.5 times better than that of the state-of-the-art 4-tap hybrid-type ToF image sensor.

## 1. Introduction

Time-of-flight (ToF) 3D image sensors have been paid much attention in various application fields. To expand the application fields of ToF image sensors for outdoor use, a sensor technology which meets the requirements of operation under direct-sunlight-level ambient light, depth range longer than 10 m, and fast response to moving objects to avoid motion artifact is necessary. Recently, the direct ToF (dToF) image sensors have gained increasing interest in various application fields, particularly in long-range light detection and ranging (LiDAR) [1,2,3,4,5,6,7,8,9,10,11]. The dToF offers excellent performance in long-range ToF measurements under direct sunlight operation by using sophisticated digital-domain processing to distinguish signal photons from ambient photons via coincidence detection of signal photons and ToF histogramming [1,10]. Because of the circuit complexity required for per-pixel processing in real time, long-range dToF image sensors for operation under direct sunlight while having a high range-image resolution will remain an issue. In contrast, another type of ToF image sensors, the indirect ToF (iToF) image sensors, have a distinct merit of high depth-image resolution because of their simple pixel configurations and simple processing for ToF measurements [12,13,14,15,16,17,18,19,20,21,22,23,24,25,26,27,28,29,30]. There are two types of light modulation and in-pixel demodulation methods in iToF: the continuous wave (CW) modulation and the pulse or short pulse (SP) modulation. CW-modulation-based iToF (CW-iToF) image sensors with two-tap demodulator pixels are most widely used because of their simple pixel configuration suitable for high image resolution and well-established performance through the long history of research and development [12]. However, the two-tap CW-iToF image sensors have issues of long-range measurement under strong ambient light and motion artifact because of the use of four subframes for depth measurements. Though techniques using 4-tap CW-iToF [24] or pseudo 4-Tap CW-iToF [27] for mitigating the motion artifact problem have been reported, these techniques need at least two subframes for long-range measurements due to the use of two modulation frequencies for unwrapping [21], leading to a motion artifact due to the timing difference of the two subframes.

To address issues on the ToF measurements under strong ambient light, SP-based iToF (SP-iToF) image sensors have a distinct advantage when compared with the CW-iToF counterparts because of the acquisition of less ambient light due to the use of a small-duty energy-concentrated light pulse in the SP-iToF [13]. The SP-iToF image sensor using the single-frame ToF calculation technique is also good for the motion artifact mitigation [17,21]. To emphasize the advantage of the SP-based iToF image sensors in long-range ToF measurements, hybrid-type SP-iToF image sensors has been proposed [31]. In the hybrid-type SP-iToF image sensors, numerous consecutive time-gating windows prepared by multi-tap pixels or the combination of the multi-tap pixels and range-shifted subframes are used for long-range ToF measurements. As a state-of-the-art development of long-range iToF-based image sensors for outdoor operations, a VGA-size 4-tap hybrid-type SP-iToF image sensor demonstrating the depth range of maximally 20 m under the ambient light level of 100 klux has been reported [29]. This sensor uses nine consecutive time-gating windows prepared by four-tap pixels and three subframes for three depth zones with the different range-shifted time-gating windows, and state-of-the-art performance at the frame rate of 15 frames/s has been demonstrated. In this technique, the motion artifact does not occur if the objects are moving within the depth range of one subframe, but there remains the concern of motion artifact if the object is moving around the boundary of two depth zones prepared by different subframes. Toward accomplishing further advanced performance in hybrid-type SP-iToF image sensors, e.g., higher frame rate, longer distance measurements, and motion-artifact-free mode using single-frame operations, the development of advanced pixel technology for realizing an increased number of taps while maintaining or improving the pixel’s fundamental performance is essential. As for multi-tap pixel technology with more than four taps, an eight-tap iToF image sensor has been reported, and SP-based ToF measurements using seven-tap and one-drain mode of operations under no ambient light condition have been demonstrated [28]. This multi-tap pixel which has a charge-storage (global shutter) structure in each tap to perform kTC noise cancelling is good for high-precision depth imaging with low signal light level and under weak ambient light operation. However, this multi-tap pixel technology is not suitable for outdoor use because of the small full well capacity design of the charge storage. 

In this work, we present new eight-tap one-drain demodulator pixel technology using multiple P-N junction diodes and its application to a hybrid-type SP-iToF image sensor developed for high-precision high-accuracy depth image measurements under strong ambient light condition. The proposed eight-tap one-drain P-N junction demodulator (PDN) uses segmented p+ hole-pinning areas of the pinned photodiode (p+/n/p- structure) as electrodes of the multi-tap demodulator [32]. Though the fundamental principle is originated from a special type of scientific CCD image sensor, PNCCD [33] or Junction CCD [34], which use the P-N junction for the one-dimensional backet-relay charge transfer, we have succeeded to design and implement the PND with a feature of high-speed two-dimensional electron transfer. The proposed eight-tap one-drain PND has a good time-response to the received short light pulse by directly modulating the electric potential of the photo-receiving area of the demodulator and high demodulation contrast to the short light pulse (width of 10 ns) is realized. On the other hand, the MOS-gate-based structures, called lateral electric field modulators (LEFMs) used for the seven-tap one-drain pixel of ref. [28] may have difficulty in achieving high-speed photocarrier response if the eight-tap one-drain demodulator is implemented, because the LEFM can control the channel electric potential of the peripheral region of photodiode only. In this work, we have succeeded to design and implement a hybrid-type Sp-iToF image sensor using the proposed eight-tap one-drain PND pixels and demonstrated the high-precision depth image measurements covering the depth range over 10 m under direct-sunlight-level ambient light based on single-frame ToF measurements. This single-frame eight-tap-based hybrid-type ToF measurement technique demonstrated for the first time in this work essentially solves the common problem of motion artifact due to the use of multiple subframes and may create new application fields of 3D sensing by using the features of motion artifact free and the potential of high range-map speed. Thanks to the improved DATA (depth-adaptive time-gating number assignment) and non-linearity error correction techniques proposed in this paper together with the eight-tap one-drain pixel technology, the developed ToF image sensor has demonstrated the best of depth linearity in over-10 m-range outdoor ToF measurements, which is 2.5 times better than the-state-of-the-art four-tap hybrid-type SP-iToF sensor [29].

The rest of this paper is organized as follows. Section 2 describes the principle and design of the eight-tap PND. Section 3 provides the proposed ToF measurement techniques. The experimental results of the implemented chip are described in Section 4. Section 5 presents the conclusion. 

## 2. Principle and Design of the 8-Tap P-N Junction Demodulator (PND) 

### 2.1. Basic PND Structure 

Figure 1 shows the basic structure and operation principle of the 2-tap version of the p-n junction demodulator (PND). In this 2-tap PND demodulator, the p+ pinning layer of the pinned photodiode (p+/n/p- structure) is segmented into multiple regions and the p+ multiple islands isolated from the surface substrate are used for gate electrodes for the channel-potential modulation. In addition to an n-type doping layer, n_1_, which is doped in the entire demodulator structure, another doping layer, n_2_, is used under the p+ gate electrodes for increasing the capacitance of each p-n junction and the resulting high potential modulation capability. In the two-tap demodulator shown in Figure 1a–c, the central gate electrode (G_C_) is used for applying a static bias and the left (G_1_) and right (G_2_)-side gate electrodes are used for dynamic control of the channel potential. The left and right-side ends of the gate electrodes, G_1_ and G_2_, are connected to n+ floating diffusions, FD_1_ and FD_2_, respectively, which are used for storages of photo carriers. For an increased sensitivity in the infrared region, a relatively thick p-epi layer on p+ substrate is used and negative backbias is applied to the backside substrate via a terminal T_SB_ for a high-speed response of carriers generated deep inside the p-epi layer. Figure 1d,e shows the horizontal potential profiles at the channel region (X_1_–X_1_’) and the surface region at the peripheral of the structure (X_2_–X_2_’). The solid line shows potential profiles when high (0.0 V) and low (−2.0 V) levels are applied at the gates, G_1_ and G_2_, respectively, to transfer photoelectrons to the left and the dashed line shows those when high (0.0 V) and low (−2.0 V) levels are applied at the gates, G_2_ and G_1_, respectively, to transfer photoelectrons to the right. The surface substrate (T_SS_) is biased to −2.0 V and the central electrode (G_C_) is statically biased to −1.0 V. As shown in Figure 1e, a sufficiently large potential barrier must be created at the gap between two adjacent p+ regions to prevent the hole current flowing between the two p+ surface regions when different terminal voltages are applied between G_1_ and G_2_ in the range of −2 V to 0 V.

On the other hand, a sufficiently large lateral electric field or large potential gradient must be created at the channel regions as shown in Figure 1d. This is necessary for high-speed lateral carrier transfer in the photo-receiving area. This basic potential modulation structure using P-N junctions is good for simultaneously having relatively large light-receiving area and attaining relatively large channel potential modulation using relatively small gate-voltage swing when compared with demodulators using transfer gates [17,19] and lateral electric-field modulation (LEFM) gates [19,21]. 

### 2.2. 8-Tap PND Pixel

The basic two-tap demodulator shown in Figure 1 can be extended to multi-tap (more than two taps) demodulators using a two-dimensional arrangement of p+ gate electrodes. Figure 2a shows the top view of the designed 8-tap demodulator using the PND structure, which has 8 p+ gate electrodes (G_1_–G_8_) for transferring photo carriers to 8 floating-diffusion regions FD_1_–FD_8_ for signal outputs, 4 p+ gate electrodes (G_D1_–G_D4_) for draining photo carriers to drains, and a central p+ gate electrodes (G_C_) for controlling the channel potential at the central region of the structure. Although there are 4 draining gates (G_D_ and 3 G_DO_), G_D_ located in-between G_8_ and G_7_ is only used for modulation because of its better draining speed and the other 3 draining gates of G_DO_ are biased to a fixed voltage. 

Figure 2b shows the pixel schematic which includes equivalent model circuits of the 8-tap demodulator with a draining gate and readout circuits using 8 source followers. The PND element is modeled by an n-channel JFET (junction field effect transistor) because the basic structure of the PND element is similar to that of the n-channel JFET. However, it should be noted that the detailed structure of the PND element is not exactly the same as that of the conventional JFET. The PND element does not have any explicit source terminal to apply bias voltage and in the entire structure of the PND other than the FD or drain terminal, electrons are fully depleted for creating drift electric field to transport a few photoelectrons to one of the FDs or drain terminals in the sub-nanosecond time range. The 8-tap demodulation pixel is driven by 8-phase short-pulse gating clocks for G_1_–G_8_ and a gating clock for the draining gate G_D_. The light pulses and gating pulses with the width of *T_0_* and cycle time of *T_C_* are repeatedly given during the signal accumulation time of *T_C_* × *N_C_*, where *N_C_* is the number of pulse cycles. After the signal-accumulation phase, the readout phase starts. The readout process from the pixel array is done on a row-by-row basis. To read out 8-tap signals using 4 vertical signal lines SO_1_, SO_2_, SO_3_, and SO_4_, the readout operation cycle is repeated twice in one row. In the first cycle, the pixel selection signal SL_1_ and the reset signal RT_1_ are used for reading the accumulated signals in FD_1_, FD_2_, FD_7_, and FD_8_. In the second cycle, the pixel selection signal SL_2_ and the reset signal RT_2_ are used for reading the accumulated signals in FD_3_, FD_4_, FD_5_, and FD_6_.

Figure 3 shows a 3D device simulation result of the designed 8-tap PND. Figure 3a shows the maximum potential profile when the generated photo-carriers are to be transferred to FD_6_. The modulation gate G_6_ is biased to a high voltage (0.0 V), and the other modulation gates G_1_–G_5_, G_7_, and G_8_ and the draining gates G_D_ and G_DO_ are biased to a low voltage (−2.0 V). The central gate G_M_ is biased to −1.0 V. The surface substrate T_SS_ and the backside substrate T_SB_ are biased to −2.0 V and −3.0 V, respectively. In Figure 3a, redpoints denote the initial positions (depth: 3 µm) of an electron generated by a photon and black thick lines indicate the movement traces of the electron. The slowest transfer time from the initial position to the site of the FD_6_, which is calculated by the velocity of carriers as a function of the electric field, is 820 ps. Figure 3d shows the maximum potential diagrams along with A–A’ line shown in the top-view layout pattern of Figure 3c, for the cases for transferring photoelectrons to G_6_ (black line) or G_2_ (red line), where G_6_ and G_2_ are biased to 0 V, respectively, and the other modulation gates are biased to −2 V. As shown in Figure 3d, the direction of carrier transfer is modulated in a wide range. At the edge of the range of carrier modulation, the channel potential is modulated in the range of −0.4 V to 1.1 V (difference of 1.5 V) by the applied gate voltage range of −2 V to 0 V. The potential modulation ratio of the channel to the gate is 75 %. This large potential modulation ratio is effectively used for the high-speed 8-tap demodulator. Figure 3e shows the maximum potential diagrams along with B–B’ line shown in Figure 3c, for the cases of transferring photoelectrons to the drain. The red solid line shows the potential plot for the case that only the G_D_ located between G_8_ and G_7_ is biased to 0 V and the other three draining gates of G_DO_ are biased to −2 V. The black solid line shows the case that four draining gates (G_D_ and 3 G_DO_) are biased to 0 V. Obviously, the case of the red solid line has a better potential profile for rapidly transferring electrons to the drain than the case of black solid line, indicating that activating the G_D_ only is better for high-speed draining.

## 3. ToF Measurements Using the 8-Tap PND Pixels

### 3.1. Depth Imaging Method Using Short Pulse and 8-Tap Pixels 

Figure 4 shows the principle of ToF measurements using a short light pulse and an 8-tap pixel with a draining gate. As shown in Figure 4a, the modulation gate pulses of G_1_–G_8_ are sequentially given from G_1_ to G_8_ and G_D_ is given complementary to the set of G_1_–G_8_ for measuring the distance from D_min_ to D_max_. The light pulse and modulation gates have the pulse width of *T_0_* and the cycle time of *T_c_*. This light pulse cycle and the gating cycle is synchronously and periodically repeated within its accumulation time to sufficiently dense the signal charge, that is, the accumulation time is given by *N_c_* × *T_c_* where *N_c_* is the number of the periodical gating cycles. Using this gate-pulse timing of the 8-tap pixel, the depth range of 0 to 7*D*_0_ can be measured, where *D*_0_ is a unit distance which can be measured by one gate pulse and is given by $D_{0}=0.5\;cT_{0}$ where *c* is the velocity of light. 

The ToF is measured with two steps based on the hybrid type of ToF measurements [29,34]. The first step of ToF measurements is to find the signal tap to which the incident light pulse is coming and to determine the coarse ToF, $T_{dC}$ as an integer multiple of *T*_0_. In this example, the coarse ToF is determined to be $T_{dC}=4T_{0}$ because the received light pulse appears at the 5th and 6th taps. The second step is to measure the fine ToF, $\Delta T_{d}$ using an SP-based indirect ToF measurement technique [21,28], where the ToF is measured with the accumulated electrons in the signal taps. When the light pulse appears at the *i*th and (*i +* 1)th taps, the fine ToF is expressed as
(1)$$\Delta T_{d}=\frac{N_{i+1}-N_{i-1}}{N_{i+1}+N_{i}-2N_{i-1}}T_{0}$$
where *N_i−_*_1_, *N_i_* and *N_i+_*_1_, are the number of signal electrons accumulated in G*_i−_*_1_, G*_i_* and G*_i+_*_1_, respectively [28]. Then, the ToF which is the sum of the coarse ToF and fine ToF is given by $T_{d}=T_{dC}+\Delta T_{d}=(i-1)T_{0}+\Delta T_{d}$. In the case of Figure 4, *i* equals to 5. 

One of difficult issues for ToF sensors to realize wide distance range is that the signal light intensity back reflected by an object largely varies as the distance of the object varies, because the irradiance at the object plane is inversely proportional to the square of distance from the ToF camera. The DATA (depth-adaptive time-gating number assignment) technique is very useful for solving this issue [28], where the best time-gating number is assigned to each gate for avoiding signal saturation for the closer objects while maintaining a sufficient signal intensity for further objects. For better cancelling of ambient light, a modified version of the DATA technique is proposed here. The timing diagram when the modulation gates of G_4_–G_8_ only are activated during the time zone for signal light sampling is shown in Figure 4b. In this example, the modulation gates of G_1_–G_3_ are activated during the time zone at very ending part of the periodical gating cycle for sampling ambient light only. Then, the ambient light is sampled in all the gates, while the signal light is sampled in G_4_–G_8_ only. An example of actual time-gating pattern using the modified DATA technique is shown in Figure 5. The number of signal sampling pulses in G_1_, G_2_, G_3_, G_4_, G_5_, G_6_, G_7_, and G_8_ is assigned to *N_c_*/8, *N_c_*/4, 3*N_c_*/8, *N_c_*/2, 5*N_c_*/8, 3*N_c_*/4, 7*N_c_*/8, and *N_c_*, respectively, where *N_c_* is the total number of the periodical gating cycles. With this technique, the same amount of ambient light is captured and the resulting same amount of ambient photocharges is accmulated in all the gates, and the calculation of the fine ToF using the modified DATA technique can be performed using Equations (1) and (2) without any modifications. 

### 3.2. Linearity and Distance Resolution Using the Modified DATA Technique 

A possible problem of the modified DATA technique is the non-linearity caused by using different number of signal sampling pulses in G_i_ and G_i+1_. This error can be explained as follows. In Equation (1), the mean values of *N_i−_*_1_, *N_i_* and *N_i+_*_1_ when the fine ToF is varying from 0 to T_0_ are expressed as $\bar{N_{i-1}}=\bar{N_{A}}$, $\bar{N_{i}}=\bar{N_{S1}}(1-x)+\bar{N_{A}}$, and $\bar{N_{i+1}}=\bar{N_{S2}}x+\bar{N_{A}}$ where *x* is a variable given by the ratio of the true fine ToF to *T_0_*, $\bar{N_{S1}}$ is the mean number of signal electrons when the light pulse is perfectly in-phase to G_i_, $\bar{N_{S2}}$ is the mean number of signal electrons when the light pulse is perfectly in-phase to G_i+1_, and $\bar{N_{A}}$ is the mean number of offset electrons due to ambient light accumulated in each tap. Then, Equation (1) as a function of *x* is expressed as
(2)$$\Delta T_{d}(x)=\frac{r_{S21}x}{1+(r_{S21}-1)x}T_{0}$$
where $r_{S21}=\bar{N_{S2}}/\bar{N_{S1}}$. Equation (2) indicates that the modified DATA technique causes non-linearity. If the modified DATA technique is not used, i.e., $\bar{N_{S1}}=\bar{N_{S2}}$, Equation (2) is simplified to $\Delta T_{d}(x)=xT_{0}$ and the fine ToF is measured with perfect linearity. By visiting the term that causes the nonlinearity in Equation (2), i.e., $r_{S21}/(1+(1-r_{S21})x)$ and knowing this function is deterministic once the operation timing of the modified DATA technique as shown in Figure 5 is fixed, one can notice that this nonlinearity can be corrected by multiplying an inverse function of it, i.e., the nonlinearity error correction is expressed as
(3)$$\Delta T_{d}{}^{\left(C\right)}=\frac{\Delta T_{d}{}^{\left(Meas\right)}}{r_{S21}{}^{\left(C\right)}}\left(1+(r_{S21}{}^{\left(C\right)}-1)\frac{\Delta T_{d}{}^{\left(Meas\right)}}{T_{0}}\right)$$
where $\Delta T_{d}{}^{\left(C\right)}$ is the fine ToF after the nonlinearity correction, $\Delta T_{d}{}^{\left(Meas\right)}$ is the measured fine ToF, and $r_{S21}{}^{\left(C\right)}$ is a constant used for error correction. The best value of $r_{S21}{}^{\left(C\right)}$ can be chosen for minimizing the nonlinearity error after correction. 

A theoretical calculation model of the precision of distance measurement using the modified DATA method can be derived by a similar way given in ref. [21,34]. Using Equation (3) and considering the difference of the number of signal sampling pulses in G_i_ and G_i+1_, the standard deviation of the distance resolution $\sigma_{D}$ in the fine ToF measurement is expressed as
(4)$$\frac{\sigma_{D}^{2}}{D_{0}{}^{2}}=\frac{1}{C_{D}{}^{2}\bar{N_{S1}}}\left(F_{S}(x)+F_{N}(x)\frac{2\left(\bar{N_{A}}+N_{R}{}^{2}\right)}{\bar{N_{S1}}}\right)$$
where $N_{R}$ is the circuit readout noise and $C_{D}$ is the demodulation contrast, and where $F_{S}(x)$ and $F_{N}(x)$ are, respectively, given by
(5)$$F_{S}(x)=\frac{r_{S21}x+r_{S21}(r_{S21}-2)x^{2}-r_{S21}(r_{S21}-1)x^{3}}{{\left(1-(1-r_{S21})x\right)}^{4}}$$
and
(6)$$F_{N}(x)=\frac{1-(2+r_{S21})x+(1+r_{S21}+r_{S21}{}^{2})x^{2}}{{\left(1-(1-r_{S21})x\right)}^{4}}.$$

If $r_{S21}=1$ or the same sampling pulse number is used in G_i_ and G_i+1_, $F_{S}(x)$ and $F_{N}(x)$ are simplified to $F_{S}(x)=x(1-x)$, and $F_{N}(x)=1-3x+3x^{2}$, respectively, which are identical to the distance precision model for the 3-tap SP-iToF sensor [21]. 

## 4. Measurement Results

### 4.1. Implemented ToF Image Sensor Chip 

Figure 6 shows the photomicrograph of the implemented ToF image sensor chip using a 0.11 µm CMOS image sensor technology. Table 1 shows a summary of specifications and basic characteristics of the chip. The implemented chip includes a 272 (H) × 60 (V) array including a 120 (H) × 60 (V) array with the 8-tap PND pixels and other arrays of test pixels in the size of 9.34 mm (H) × 7.04 mm (V). The 9-phase gating clock signals for driving G_1_ through G_8_ and G_D_ are supplied from outside FPGA and are given to the 8-tap PND pixel array through a driver. Using 4 parallel signal lines per pixel (22.4 µm pitch) and a column ADC [35] with pitch of 5.6 µm, 4 of 8 outputs from each pixel are read out and converted to 16 bit digital signals in parallel. One row of pixel signals each of which is half of 8-tap pixel outputs is horizontally scanned and read out using an LVDS (low-voltage differential signaling) interface. The column ADC used is the folding integration/cyclic ADC using a multiple sampling technique [35]. The readout time of the one row of the pixel signals takes 80 μs including the times required for the analog-to-digital conversion and digital CDS. The total readout time for reading one frame image signal, i.e., the time reading 120 rows, each of which consists of two sets of 4 parallel signals from the 8-tap pixel, is 9.6 ms (=120 × 80 μs). The input-referred noise of the ADC is 130 μV_rms_, which is sufficiently low compared with the noise originated by the pixel behavior of the designed chip. A relatively low conversion gain (measured: 10.8 μV/e^−^) is used in order to have a sufficiently high full well capacity (measured: 79.2 ke^−^). The fill factor defined by the ratio of the area of demodulation to pixel size is 9% and using a microlens, the quantum efficiency is 14.5% at 940 nm. The read noise of the major component is the kTC noise 745 μV_rms_, or 69 e^−^_rms_. 

The dark current of the 8-tap demodulator pixel is measured at 60 °C using one of the 8-tap outputs (the 6th tap) and two different measurement conditions: One is the static gate voltage setting of G_6_: High, G_D_: Low, and the other gates (G_1_–G_5_, G_7_, G_8_): Low, and the other is the dynamic gate driving, as shown in the timing diagram of Figure 4. In the dynamic gate driving, the gates G_6_ and, G_D_ are pulsed with T_0_ = 10 ns and T_C_ = 250 ns, and the other gates (G_1_–G_5_, G_7_, G_8_) are statically set to Low. In the static gate voltage setting, the dark current generated in all the aeras of the demodulators are captured in the FD of the 6th tap and the dark current in this case is 506 pA/cm^2^ or 15.9 ke^−^/s with 22.4 µm pixel. In the dynamic gate driving, the dark current is reduced to 166 pA/cm^2^ or 5.6 ke^−^/s because the draining gate is opened during 96% of the gating cycle time and the dark current generated in most of the demodulator areas are drained out.

### 4.2. Response to Light Pulse Delay 

Figure 7 shows measurement results of the response of 8-tap outputs to the light pulse delay. In Figure 7a,b which is the normalized output of Figure 7a, a laser with the pulse width of 10 ns and the wavelength of 940 nm is used to characterize the demodulation contract. A lens with a focal length of 12.5 mm is used for the sensor characterization. In this measurement, the 8-tap PND gates, G_1_ through G_8_ are opened with *T_0_* of 10 ns as shown in the timing chart of Figure 4a, and the delay of the laser pulse is scanned from 0 to 100 ns with a step of 0.3 ns. As shown in Figure 7b, the response of the eight signals to the pulse delay time is fast enough for the short-pulse ToF measurement with the light pulse width of 10 ns when compared with the ideal response of triangular shape. The distortion of the response from the ideal response, denoted by the demodulation contrast, will influence the depth precision of the fine ToF measurements using Equation (3). For example, if the received light pulse appears at the second and third taps, the fine ToF is measured using the tap signals of G_1_, G_2_, and G_3_. Then, the demodulation contrast defined by $\max(G_{2}/(G_{1}+G_{2}+G_{3}))\times 100[\%]$ in this fine ToF measurement is calculated to be 88% from the response curve of Figure 7b. Figure 7c shows a response curve of the 8-tap pixel to a very-short pulse laser for characterizing the carrier response speed of the 8-tap demodulator. A semiconductor pulse laser with the FWHM (full width half maximum) of 69 ps, the wavelength of 851 nm, and the peak power of 88 mW is used for the light source and the delay of the laser pulse is scanned from 0 to 110 ns with a step of 0.3 ns. Figure 7d,e show the time derivative of the outputs of Figure 7c showing the carrier response curves at the rising and falling edges of 8 gating time windows, and the FWHM measured with the response curves, respectively. The FWHM is ranging from the worst of 3.45 ns in G1 and the best of 2.0 ns in G7, and the mean value is 2.4 ns.

### 4.3. Range Measurement Performance

Figure 8 shows the experimental setup for ToF measurements under ambient-light illumination. In this measurement, a white reflector board mounting an ambient light illuminator with the wavelength of 940 nm is used and the distance from the reflector board to the ToF camera is changed by moving the reflector board which is mounted on a mobile cart. The F-number setting of the lens is 4 and a NIR band-pass filter with the passband of 950 + 25/−25 nm is used for the lens. The intensity of the ambient-light illuminator is set at the equivalent illuminance of 80 klux. By mounting the ambient-light illuminator on the reflector board, the ambient-light intensity at the reflector board plane is kept unchanged while changing the distance of the reflector board to the ToF camera. The pulse width of the emitted laser light and the gating pulse width used in the pixel is 10 ns. The reflector board has Lambertian surface with the reflectance of 99%. The DATA method described in Section 3 is used and the accumulation times in the first through eighth taps are set to 0.500 ms, 2.400 ms, 5.500 ms, 9.250 ms, 14.00 ms, 19.75 ms, 26.00 ms, and 33.25 ms, respectively. The data points are measured by changing the distance of the reflector board from the ToF camera in the range from 1.0 m to 11.5 m with the step of 0.25 m. Figure 9a shows the measurement results of distance and its linearity using the 8-tap ToF imager under the ambient light illumination of 80 klux. Figure 9a shows the distance measurement results and the nonlinear error. The maximum non-linearity error is 1.4% of the full-scale range of 11.5 m. Figure 9b shows the distance measurement results and the non-linearity errors when the non-linearity error correction technique described in Section 3.2 is used. The non-linearity in the closest (the first zone) and furthest zone (the seventh zone) only is corrected by using Equation (3), and $r_{S21}{}^{\left(C\right)}$ in the first zone and the 7th zone are 2.0 and 1.4, respectively. With this non-linearity compensation, the maximum non-linearity is reduced to 0.6% of the full-scale range.

Figure 10 shows the measured depth precision for the operation under strong ambient light and dark condition, and the comparison with the theoretical calculation results using Equation (4). In the measured depth range of 1.0 m to 11.5 m, the depth precision is maximally 16.4 cm and 6.9 cm for the measurements under ambient light (80 klux) and dark, respectively, corresponding to the relative depth precision of 1.4% and 0.5% to the full-scale range, respectively. The detailed behavior of the depth precision in the 7 zones is in good agreement with the theoretical prediction given by Equation (4). The number of signal-light electrons and background-light electrons, which are necessary for plotting the theoretical curves using Equation (4), is determined by their measurements at the distance of 9.0 m.

Figure 11 shows the measured depth images used for the depth non-linearity and depth precision measurements in Figure 9 and Figure 10. These depth images are taken while moving the reflector board in the range from 1.0 m to 11.5 m with the step of 0.25 m. A strong light equivalent to 80 klux of sun light is illuminated to the reflector board with the infrared LEDs attached to the reflector board as shown in Figure 8. It is demonstrated also from the set of depth images shown in Figure 11 that the depth measurement under sunlight-level ambient light is successfully conducted using the implemented ToF imager with the 8-tap PND pixels. 

Table 2 shows a comparison of this work with other iToF image sensors. The important feature of this work is that long range (>10 m) measurement under high ambient light (80 klux) using single subframe operation is demonstrated. This feature comes from the proposed SP-modulation based hybrid type of ToF measurements using an aggressive 8-tap pixel design and the improved DATA technique. Though the ToF image sensor in Ref. [28] also has demonstrated the single subframe operation using a similar 8-tap (more exactly, 7-tap and 1-drain because of the special usage of one of the 8 taps as a drain) pixel architecture, the operation under high ambient light is not shown. The sensor of the ref. [28] has charge storages (global shutter structure) with relatively small full-well capacity (FWC) at each tap for low-noise readout using true-CDS operation. Because the small FWC design and the original DATA technique are not suitable for high ambient light operation, the ToF image sensor of Ref. [28] has demonstrated high precision for no ambient light condition only. Because of the remaining problem of depth nonlinearity caused by the DATA technique, the depth nonlinearity is 1.56% compared to the full-scale range. The depth non-linearity measured under strong ambient light in this work is 0.6% after the error correction using Equation (3), which is better than 1.5% of ref. [29] and 1.56% of ref. [28].

In general, high range-image resolution is one of the common advantages of iToF image sensors when compared with dToF counterparts, particularly if the target application is for outdoor use. The pixel size of 22.4 µm × 22.4 µm in this work is large compared with other iToF imagers, but it still has an advantage in depth imaging pixel size when compared with one of the state-of-the-art dToF imagers for LiDAR (light detection and ranging) applications, where the depth imaging pixel size is 30 µm × 30 µm [10]. However, the shrinkage of the pixel size in the proposed 8-tap pixel design while maintaining the pixel’s fundamental characteristics is very important as a future work. The area for a charge draining structure in the present 8-tap 4-drain PND demodulator can be reduced by reducing the number of drain electrodes without degrading the demodulation characteristics. The present PND structure needs a relatively large isolation space between the tap electrodes for reducing the hole leakage current. The isolation space can be reduced by applying a modification of process steps without increasing the hole leakage current. The use of an advanced process node will help in reducing the areas of both the demodulator and readout circuits.

## 5. Conclusions

In this paper, we have presented a time-of-flight (ToF) image sensor using eight-tap p-n junction demodulator (PND) pixels. Using the proposed PND pixel structure, we have succeeded to design and implement an eight-tap one-drain demodulator with large photosensitive area while realizing a high-speed eight-phase demodulation and high-speed charge draining. In contrast to the MOS-gate-based design used for a seven-tap one-drain demodulator where the MOS gate can modulate the electric potential of peripherals of photosensitive area only, the PND can modulate the electric potential of the entire photosensitive area, and therefore we have successfully developed the eight-tap one-drain demodulator with high-speed photocarrier response for the first time. Using the eight-tap one-drain PND in each pixel and the improved DATA (depth-adaptive time-gating-number assignment) technique proposed in this work, which allows to measure the depth with well-balanced precision in the range from near to far and has the capability of ambient-light cancelling, the implemented ToF image sensor prototype works at a wide depth range of 1.0 to 11.5 m with the high depth precision (depth noise) of better than 1.4% of the maximum range under direct-sunlight-level ambient light of 80 klux. Thanks to the hybrid-type ToF measurements and by applying the proposed non-linearity correction technique, a good depth measurement nonlinearity of 0.6% of the full-scale range has been demonstrated. This linearity realized in over 10 m range and high ambient light conditions is 2.5 times better than the-state-of-the-art hybrid-type ToF sensor reported. Another finding in this work is that the good precision and linearity are realized by using the eight-phase demodulation and single-frame signals only. This single-frame eight-phase hybrid-type ToF measurement technique demonstrated for the first time in this work solves the problem of motion artifact in ToF image sensors when using multiple subframes for ToF measurements and leads to a motion-artifact-free and high-frame-rate 3D imaging. A possible issue in the current design of the proposed pixel technology is its large pixel size. For demonstrating the feature of high range-image resolution in the proposed ToF sensors, the shrinkage of the pixel size while maintaining the basic performance of ToF sensors for outdoor use will be necessary, which is left as a future subject.

## Figures and Tables

**Figure 1 sensors-23-03987-f001:**
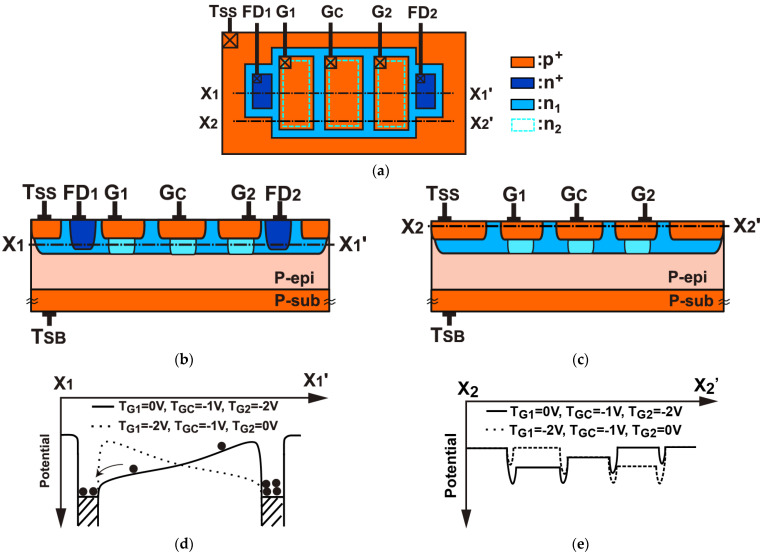
Structure and principle of the two-tap p-n junction demodulator (PND): (**a**) Top view; (**b**) Cross-sectional view (X_1_–X_1_’); (**c**) Cross-sectional view (X_2_–X_2_’); (**d**) Potential diagram at the channel (X_1_–X_1_’); (**e**) Potential diagram at Si surface (X_2_–X_2_’).

**Figure 2 sensors-23-03987-f002:**
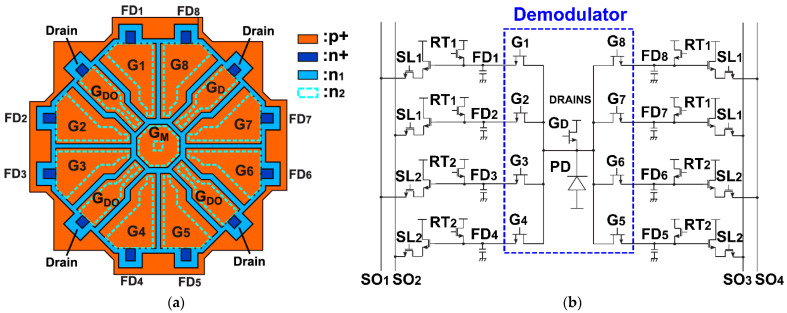
8-tap demodulation pixel and the operations: (**a**) Top view of the 8-tap PND; (**b**) equivalent pixel readout circuits.

**Figure 3 sensors-23-03987-f003:**
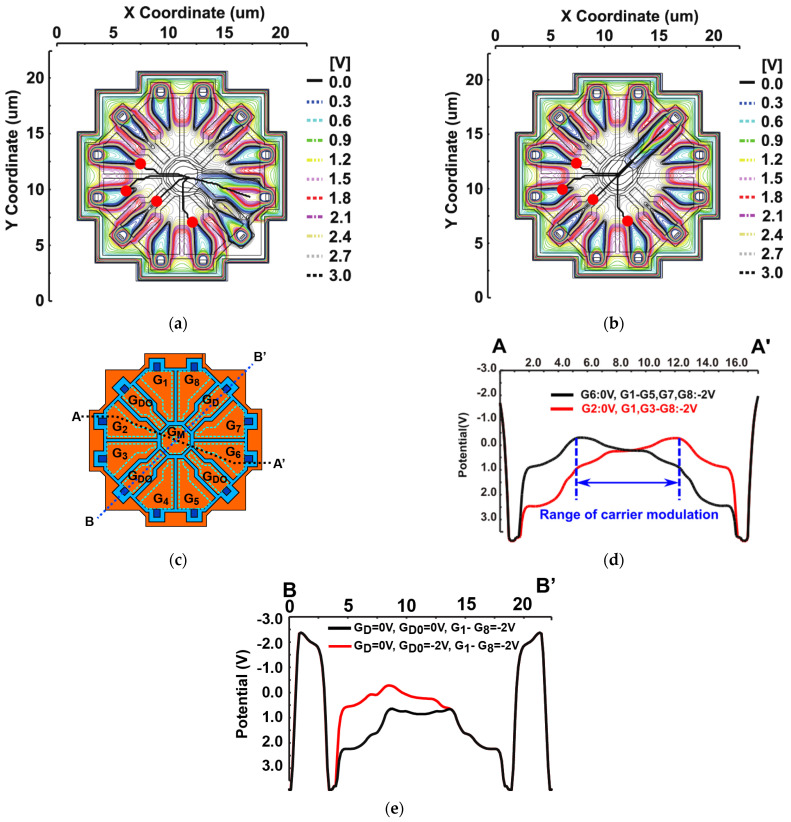
3D device simulation results of the 8-tap PND: (**a**) X-Y 2D potential plot and carrier traces to transfer to G_6_; (**b**) X-Y 2D potential plot and carrier traces to transfer to G_D_; (**c**) demodulator top view; (**d**) 1D potential plot (A–A’) for carrier transfer to floating diffusions, FD_6_ and FD_2_; (**e**) 1D potential plot (B–B’) for carrier transferring to a drain through G_D_ only (red line) and that for carrier transferring to a drain through G_D_ and G_DO_ (black line).

**Figure 4 sensors-23-03987-f004:**
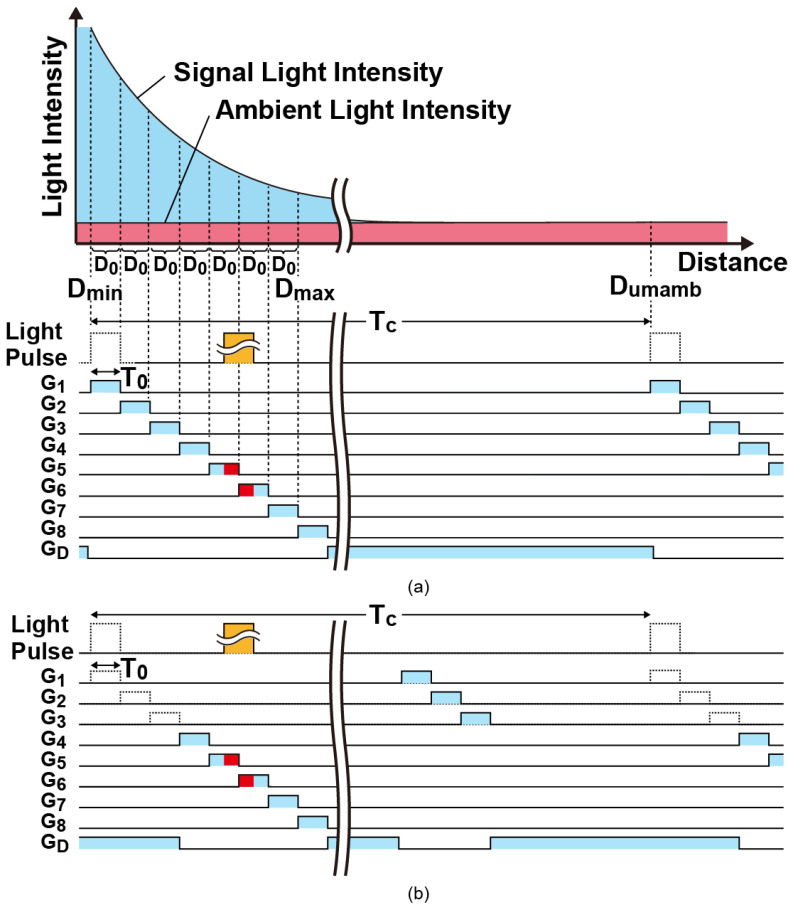
Gate timing and its correspondence to the depth range to be measured: (**a**) Gate timing when all the gates are activated in every cycle and its correspondence to the distance profile of the back-reflected light intensity; (**b**) Gate timing when G_4_–G_8_ are activated for signal light sampling and G_1_–G_3_ are activated for ambient light sampling.

**Figure 5 sensors-23-03987-f005:**
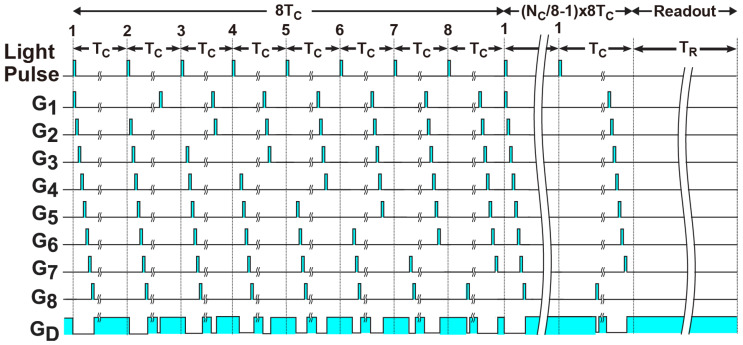
Example of the modified DATA timing diagram for cancelling ambient light.

**Figure 6 sensors-23-03987-f006:**
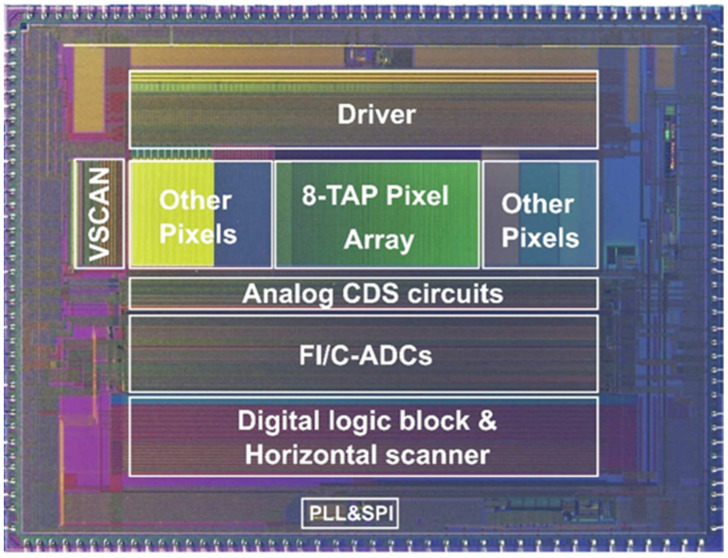
Chip micrograph.

**Figure 7 sensors-23-03987-f007:**
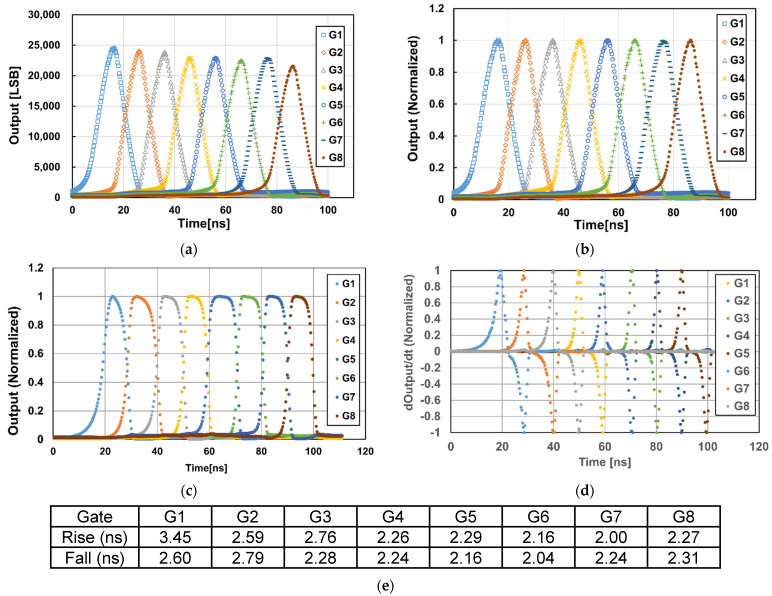
Response of the 8-tap outputs to the light pulse delay. (**a**) Response to Short Pulse (940 nm, T_0_ = 10 ns). (**b**) Response to Short Pulse (T_0_ = 10 ns, Normalized). (**c**) Response to Very Short Pulse (FWHM = 69 ps, 851 nm, Normalized). (**d**) Time Derivative of (**c**) by The Delay Time (Normalized). (**e**) FWHM of The Pixel Response to Very Short Pulse (FWHM = 69 ps) Measured with (**d**).

**Figure 8 sensors-23-03987-f008:**
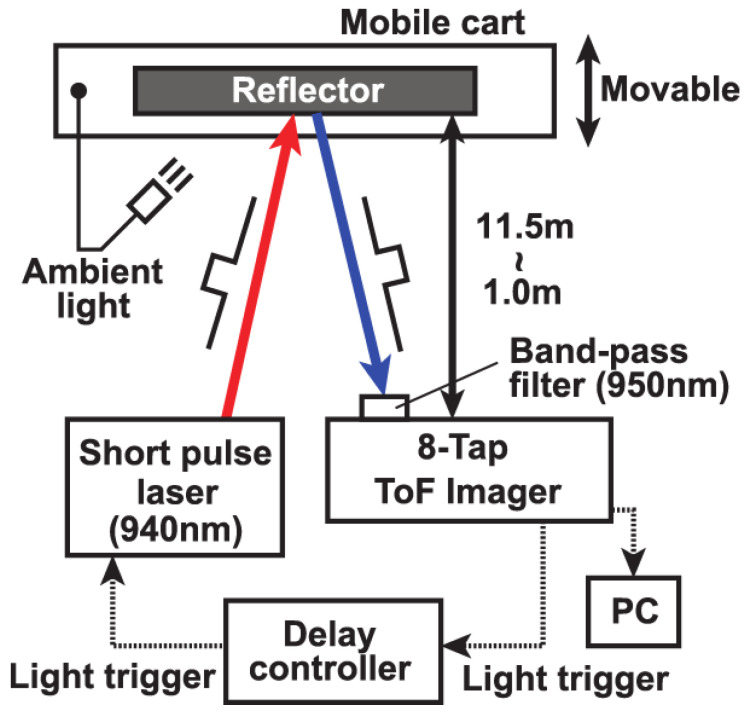
Experimental setup for ToF measurements under ambient-light illumination.

**Figure 9 sensors-23-03987-f009:**
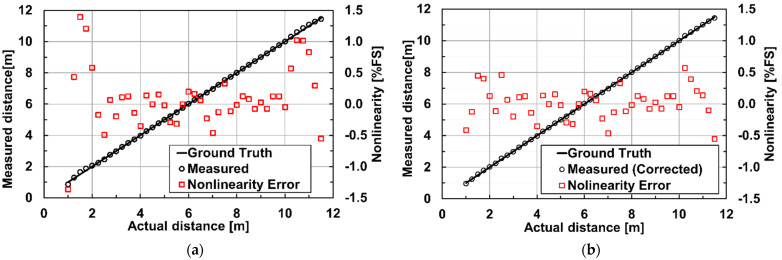
Measurement distance linearity and nonlinearity errors under background light conditions: (**a**) measured distance and non-linearity error (without error corrections); (**b**) measured distance and non-linearity error (with error corrections).

**Figure 10 sensors-23-03987-f010:**
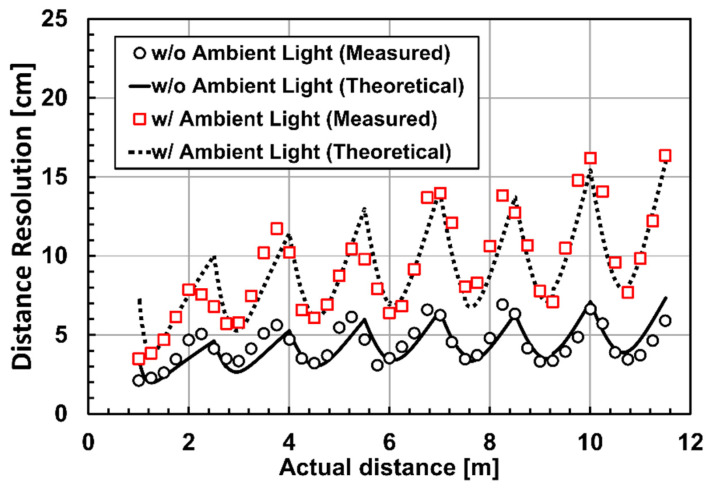
Measured and theoretical depth precision under ambient light illumination and dark for the distance range of 1.0 m to 11.5 m.

**Figure 11 sensors-23-03987-f011:**
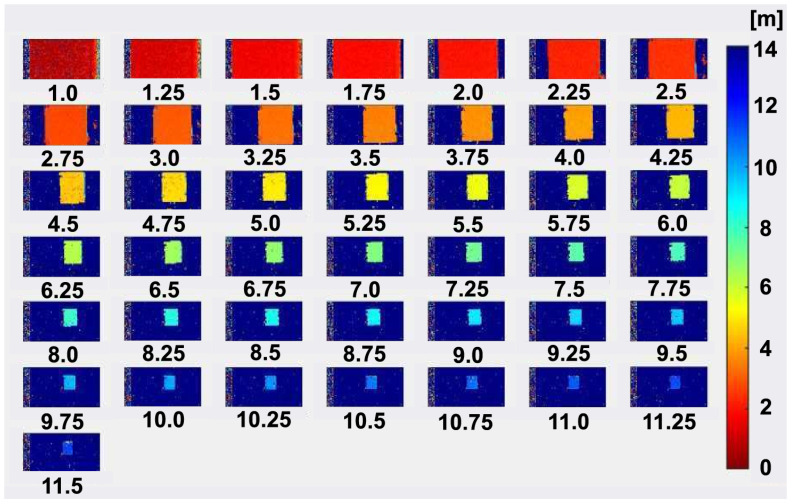
Depth image (1.0 m to 11.5 m) while moving a reflector board.

**Table 1 sensors-23-03987-t001:** Summary of the prototype CMOS imager performance.

Parameter	Value
Technology	0.11 µm CMOS Image sensor
Number of Pixels	272(H) × 60 (V) (Entire Array) 120 (H) × 60 (V) (8-Tap PND Pixel Array)
Pixel size	22.4 µm × 22.4 µm
Chip size	9.34 mm × 7.04 mm
ADC resolution	16 bit
Readout time	9.6 ms (16 bit)
Conversion gain	10.8 µV/e^−^
Full well capacity	79.2 ke^−^
Fill factor	9.0%
Quantum Efficiency Read noise Dark Current @60 °C	14.5% (940 nm) 69 e^−^ 506 pA/cm^2^ (15.9 ke^−^/s) [G_6_: High, G_D_: Low] 166 pA/cm^2^ (5.6 ke^−^/s) [G_6_, G_D_: Pulsed]

**Table 2 sensors-23-03987-t002:** Comparison of specifications and performance.

	This Work	Ref. [28]	Ref. [22]	Ref. [26]	Ref. [27]	Ref. [29]
Process	110 nm FSI	110 nm FSI	65 nm BSI	65/90 nm BSI	90 nm BSI	110 nm BSI
Pixel pitch	22.4 µm	22.4 µm	3.5 µm	3.5 µm	8.0 µm	5.6 µm
Pixel Architecture	8-tap 1-drain	7-tap 1-drain	2-tap	2-tap	Pseudo 4-tap	4-tap 1-drain
Storage (Global Shutter)	No	Yes	Yes	No	No	No
Light Modulation	SP	SP	CW	CW	CW	SP
Image Resolution	120 × 60	134 × 128	1024 × 1024	1280 × 960	320 × 240	640 × 480
Number of Subframes	1	1	4	4	4	3
Ambient Light Tolerance	80 klux	n.a.	25 klux	80 klux (QVGA)	130 klux	100 klux
Depth Range	1–11.5 m	1–6.4 m	0.4–4.2 m	1–10 m	0.75–4 m	0.5–20 m (1–30 m@0 lux)
Precision (Depth noise)	1.4%@11.5 m (80 klux) 0.6%@11.5 m (0 lux)	0.24%@6.4 m (0 lux)	0.7%@4 m (25 klux)	1.6%@10 m (80 klux)	0.54%@4 m (130 klux)	1.3%@1–20 m (100 klux)
Max. Depth Nonlinearity	0.6%@1–11.5 m (80 klux)	1.56%@1–6.4 m (0 lux)	0.05%@0.25–4 m (25 klux)	-	-	1.5%@1–20 m (100 klux)

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
