# Peer review of "A Time-of-Flight Image Sensor Using 8-Tap P-N Junction Demodulator Pixels"

_sensors, 2023, doi:10.3390/s23083987_

Round 1

Reviewer 1 Report

This paper presents an 8-tap SP-based hybrid-type ToF image sensor developed for high-precision high-accuracy depth image measurements under strong ambient light condition. The ToF image sensor chip is implemented using a 0.11 μm CMOS  and measured. The paper can be accepted after minor revision. Some comments are listed as follows:

1-The Image Resolution of the proposed design  is low  but in the text it is mentioned  that its is suitable for high-precision high-accuracy depth image measurements. Add explanations about this issue.  

2-Most part of paper are copied from [R1] without any changes which should be modified.

3- Add [R1] in the comparison table and compare proposed design with this paper. Also provide novelty of the paper compared with [R1].

4- In Fig 7, which Response of the 8-tap outputs to the light pulse delay is depicted provide the normalized data curve.

5- Equations which are not belong to authors should be cited.

[R1] Shirakawa Y, Yasutomi K, Kagawa K, Aoyama S, Kawahito S. An 8-tap CMOS lock-in pixel image sensor for short-pulse time-of-flight measurements. Sensors. 2020 Feb 14;20(4):1040.

Author Response

Dear Reviewer, 

The response to your review is uploaded as a Word file. 
For your reference, the Reply (Summary of Revisions) includes  replies to all the reviewer's comment. 
Please check the section of [From Reviewer 1] and please briefly visit to the sections to other reviewers too. 

Thank you for your reviewing. 

Best Regards, 

Authors

Reviewer 2 Report

Comments

In the current manuscript “A Time-of-Flight Image Sensor Using 8-Tap P-N Junction Demodulator Pixels”, the author has claimed to fabricate the time-resolved image sensor based on 8 Tap PN junction demodulator pixels for short-pulse time-of-flight measurements. The design is implemented with multiple PN junctions used for modulating the electric potential to transfer photoelectrons with the advantage of high-speed demodulation in large photosensitive areas. The pixel is suitable for long-range short-pulse-based time-of-flight measurements under high ambient light with single subframe operations.   The research work on time-of-flight image sensors is interesting for optoelectronic commercial applications. There are several points required to be addressed for acceptance. The manuscript can be accepted after the following revisions.

1.               English and grammar mistakes should be improved. The complete word for the abbreviation should be mentioned in the first place.

2.               The abstract should be revised using professional and technical terms. The author should provide more details of the key achievement parameters for this research work.

3.               The introduction should be revised with properly specified information and story flow. Please add the latest research progress on optoelectronic devices such as https://doi.org/10.1016/j.surfin.2022.101772. The last paragraph of the introduction should explain briefly how your work will add innovation. The size of the paragraphs should be almost the same for the whole manuscript.

4.               Please remove the unnecessary equations and cite the reference if they are available in the literature.

5.               Please revise Figure 9 and Figure 10 with a better visual view for the readers. Choose a suitable color combination and keep the font and text format symmetrical for the graphs of the whole manuscript.

6.               Please replace Figures 1,2, 3, 4, and Figure 5  with High-quality images.

7.               Please revise the conclusion by providing more concise and compact findings in this research work.

Author Response

Dear Reviewer, 

The response to your review is uploaded as a Word file. 
For your reference, the Reply (Summary of Revisions) includes  replies to all the reviewer's comment. 
Please check the section of [From Reviewer 2] and please briefly visit to the sections to other reviewers too. 

Thank you for your reviewing. 

Best Regards, 

Authors

Reviewer 3 Report

Reviewer comments

Authors presented a time of flight image sensor, using a 8-tap PN junction demulator pixels in this work.

In general, article is well-written and presents an organized scheme. However, some comments are listed below.

1.      Line 125. Fig. 1 caption is not correctly written or Figure 1. Cont must be reconsidered in order to improve readability.

2.      Line 143. PND element is modeled by a JFET approach. Please describe the characteristics considered for the JFET model such as Vknee, Vpinch-off, and drain-source saturation current in order to extend the comprenhension on your considerations.

3.      Line 290. Readout time is presented without tolerance nor variance analysis to ensure repeatibility and consistency among measurements. In general, the methodology for validation of results are nor mentioned in the manuscript for noise-sensitive monitoring system.

4.      Figure 7. Y-axis should be normalized in order to only compare the time offset of the 8-tap output. FWHM for each response could be adressed in order to remark the consistency along the 8-tap outputs.

5.      Figure 7. Caption should be revised and rewritten.

6.      Line 347. Maximum non-linearity error is 1.4% (0.6% after compensation), a comparative to other results in the state-of-the-art must be performed and included in the manuscript, considering the good-agreement with experimental results in Fig. 10.

7.      Conclusion should be more realistic, comparing results with the state-of-the-art in order to consider, as mentioned, future works.

Author Response

Dear Reviewer, 

The response to your review is uploaded as a Word file. 
For your reference, the Reply (Summary of Revisions) includes  replies to all the reviewer's comment. 
Please check the section of [From Reviewer 3] and please briefly visit to the sections to other reviewers too. 

Thank you for your reviewing. 

Best Regards, 

Authors

Round 2

Reviewer 3 Report

The authors answered almost all of my observations.